# Carotenoids and Their Interaction with the Immune System

**DOI:** 10.3390/antiox14091111

**Published:** 2025-09-12

**Authors:** Miguel Medina-García, Andrés Baeza-Morales, Pascual Martínez-Peinado, Sandra Pascual-García, Carolina Pujalte-Satorre, Rosa María Martínez-Espinosa, José Miguel Sempere-Ortells

**Affiliations:** 1Immunology, Cellular and Developmental Biology Group, Department of Biotechnology, University of Alicante, Ap. 99, E-03080 Alicante, Spain; miguel.medina@ua.es (M.M.-G.); andres.baeza@ua.es (A.B.-M.); sandra.pascual@ua.es (S.P.-G.); cps58@alu.ua.es (C.P.-S.); josemiguel@ua.es (J.M.S.-O.); 2Biochemistry and Molecular Biology and Edaphology and Agricultural Chemistry Department, Faculty of Sciences, University of Alicante, Ap. 99, E-03080 Alicante, Spain; rosa.martinez@ua.es; 3Applied Biochemistry Research Group, Multidisciplinary Institute for Environmental Studies “Ramón Margalef”, University of Alicante, Ap. 99, E-03080 Alicante, Spain

**Keywords:** phytochemicals, nutraceuticals, antioxidant, immunomodulation

## Abstract

Carotenoids are lipophilic pigments naturally occurring in plants and, to a lesser extent, in certain non-photosynthetic organisms. They play a critical role in human health due to their antioxidant and immunomodulatory properties. Key carotenoids such as β-carotene, lycopene, lutein, and zeaxanthin are capable of neutralizing reactive oxygen species, thereby mitigating oxidative stress—a major contributor to the onset and progression of chronic diseases. These compounds also modulate immune responses by influencing lymphocyte proliferation, enhancing natural killer cell activity, and regulating the production of pro- and anti-inflammatory cytokines. Such immunomodulatory effects are associated with a reduced risk of infectious diseases and have shown potential protective roles against inflammatory conditions, cardiovascular and neurodegenerative disorders, and certain types of cancer. Moreover, diets rich in carotenoids are linked to improved immune status, particularly in vulnerable populations such as the elderly and immunocompromised individuals. Despite strong epidemiological evidence, clinical trials involving carotenoid supplementation have produced mixed results, indicating that their effectiveness may depend on the broader dietary context and interactions with other nutrients. In summary, carotenoids are important dietary compounds that contribute to immune regulation and the prevention of various diseases, although further clinical research is needed to determine optimal intake levels and assess their full therapeutic potential.

## 1. Introduction

### 1.1. Overview of Carotenoids

Carotenoids are organic compounds formed by the polymerization of 2-methyl-1,3-butadiene, commonly called isoprene, typically consisting of eight units and totaling 40 carbon atoms [1]. However, variations include nor-carotenoids (with removed atoms), apo-carotenoids (shortened skeletons), diapo-carotenoids (C_30_), and higher carotenoids (C_45_, C_50_) [2,3]. The stereoisomerism of the final molecule allows double bonds in both cis and trans configurations, resulting in numerous geometric isomers of carotenoids [4]. It is important to note that carotenoids are lipophilic molecules, meaning that they are typically located in the least hydrophilic regions of cells. However, they can undergo esterification with fatty acids, which reduces their polarity once esterified [5].

Carotenoids are primarily found in plants, performing critical functions such as light capture and photoprotection [6], but non-photosynthetic microbial organisms can also produce them for defense against photo-oxidative damage in light-exposed, oxygen-rich environments [7]. For example, microalgae serve as primary producers of astaxanthin, lutein, and β-carotene [8]; and fungi primarily produce β-carotene, torularhodin, torulene, and astaxanthin [9]. However, while superior organisms, like humans, cannot synthesize carotenoids, they rely on dietary intake to benefit from their antioxidant properties and maintain normal physiological functions [10].

Since plants are the most common source of carotenoids, the carotenoid biosynthetic pathway has been extensively studied in numerous plant species. In recent years, significant attention has been given to the regulatory control of carotenoid metabolism [11,12,13,14]. However, carotenoid degradation and storage are also key areas of research interest [14,15,16].

### 1.2. Classification of Carotenoids

Carotenoids can be classified into two distinct groups based either on their chemical composition or their molecular structure (Figure 1).

Based on chemical properties, xanthophylls, such as lutein, zeaxanthin, and β-cryptoxanthin, are characterized by at least one oxygen-containing functional group, while carotenes, including β-carotene, α-carotene, ζ-carotene, phytoene (PT), lycopene, and phytofluene (PTF), are purely hydrocarbons. Based on their molecular structure, specifically attending to the terminal rings, this classification includes monocyclic (e.g., δ-carotene and γ-carotene), bicyclic (e.g., α-carotene, β-carotene, β-cryptoxanthin, lutein, and zeaxanthin), and acyclic carotenoids (e.g., PT, PTF, lycopene, and ζ-carotene) [17]. Additionally, a representative example of each classification is shown in Table 1.

**Table 1 antioxidants-14-01111-t001:** Representative examples of carotenoids of both types of classification.

Based on chemical composition	Xanthophyls	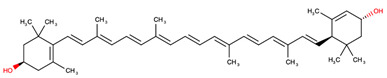 lutein
Carotenes	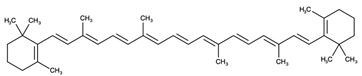 β-carotene
Based on molecular structure	Monocyclic	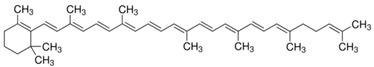 γ-carotene
Bicyclic	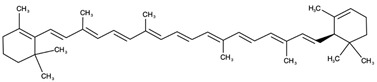 α-carotene
Acyclic	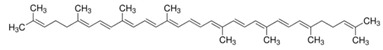 lycopene

### 1.3. Applications of Carotenoids in Health

Although carotenoids, like β-carotene and α-carotene, were primarily recognized for being a provitamin A source, essential for vision and the immune system [18], they are not classified as essential nutrients, and, thus, not directly linked to a recommended daily intake value [19]. However, contemporary scientific evidence has revealed a multitude of potential health benefits associated with the consumption of carotenoids. For example, there is a correlation between carotenoid intake and a reduced risk of various diseases, including cancer [20], cardiovascular [21], degenerative, autoimmune [22], and eye diseases [23]. Among their main attributes is their antioxidant capacity; other properties must be highlighted, including regulation of intracellular signaling pathways [24], pro-oxidant activity and light absorption [25], modulation of membrane properties, and various aspects associated with immune system modulation [5] (Figure 2). Consequently, the relationship between carotenoid presence and the incidence of these, and other, health conditions continues to be a highly compelling area of research [21].

## 2. Carotenoids and Their Relationship with the Immune System

### 2.1. Antioxidant and Anti-Inflammatory Activity of Carotenoids

Carotenoids have been studied for their immune-boosting, anti-inflammatory, and antioxidant effects in both in vitro and in vivo studies [26]. In the first place, β-carotene reduces in vitro oxidative stress by lowering pro-inflammatory adipokines, like MCP-1 and RANTES, and increasing adiponectin in 3T3-L1 adipocytes after pretreatment with 5 µM or 10 µM β-carotene [27]. It also inhibits NF-κB activation and IL-8 expression triggered by hydrogen peroxide in gastric epithelial AGS cells after in vitro pretreatment with 20 µM of synthetic β-carotene [28], and it suppresses iNOS and COX-2 expression in cells infected with *Helicobacter pylori* at concentrations between 2 and 20 µM [29]. Furthermore, synthetic β-carotene (10–100 µM) may reduce the phosphorylation of JAK2 and STAT3, suggesting that its anti-inflammatory effect could be mediated through the inhibition of the JAK2/STAT3 signaling pathway, and it also reduces the phosphorylation of NF-κB, JAK2, STAT3, JNK, and p38 in murine macrophage cell line RAW264.7 and murine peritoneal macrophages obtained from BALB/c mice [30], which are activated by LPS [31,32].

Astaxanthin (3,3-dihydroxy-β, β-carotene-4,4-dione) (ASTA) supplementation has been shown to decrease lipid peroxidation and protect against cellular damage in healthy volunteers for four weeks at a daily dose of 3 mg in softgel capsules in a study with ASTA obtained from *Haematococcus pluvialis* [33]. Studies also indicate that carotenoids with electrophilic groups can interact with NF-κB subunit cysteine residues, inhibiting their activation and promoting the elimination of ROS in cancer cells in vitro [34]. Synthetic ASTA exhibits anti-inflammatory effects in vitro in THP-1 macrophage cells pretreated with 5–10 µM by inhibiting NF-κB activation and downregulating pro-inflammatory markers such as IL-1, IL-6, TNF-α, MMP-2, and MMP-9 [35]. In addition, it can reduce p-PI3K, p-AKT, and p-mTOR levels within *H. pylori*-infected cells after pretreatment with 1–10 µM [36]. Moreover, in a study on smokers, supplementation with *Haematococcus*-derived ASTA for three weeks at daily doses between 5 and 40 mg resulted in a dose-dependent reduction in F2-isoprostane levels, a lipid marker of oxidative stress [37].

Crocin and crocetin from saffron also display a wide range of pharmacological effects. Crocin, derived from the stigma of Crocus sativus, possesses strong antioxidant properties [38]. The neuroprotective effects of crocin and crocetin are largely attributed to their antioxidant activities [39]. Crocin from *Crocus Sativus* L. (10 µM in serum-free DMEM) increases intracellular glutathione, protecting hypoxic PC12 cells (a model for brain ischemia) from cell death [39]. Synthetic crocetin, for instance, suppresses mRNA expression of TNF-α, IL-1β, and iNOS in the liver, improving survival in Sprague–Dawley male rats (hemorrhagic shock model) after 1 mL/kg administration [40]. Both synthetic crocin and crocetin inhibit nitric oxide release induced by LPS after pretreatment in BV2 mouse brain microglial cells and reduce NF-κB activation, and TNF-α, IL-1β, and intracellular ROS levels. These compounds exhibit neuroprotective potential by lowering neurotoxic molecule production from activated microglia, a response to interferon-gamma (IFN-γ) and beta-amyloid (Aβ) [41]. In multiple sclerosis (MS) research, crocin was shown to impact inflammatory oxidative markers significantly. Saffron’s anti-inflammatory and antioxidant properties reduce oxidative stress markers and inflammation, and in a 4-week treatment with crocin, lipid peroxidation (LPO) decreased, as well as DNA damage, while increasing total antioxidant capacity (TAC) and total thiol groups (TTG) levels compared to a placebo group [22]. Studies also support the potential of saffron in neurodegenerative conditions by mitigating oxidative stress and inflammation [42]. One study found that synthetic crocin administered at 30 or 60 mg/kg per day for 6 weeks improves memory in Parkinsonian rats (Adult male Wistar rats) by reducing nitrate and thiobarbituric acid reactive substances (TBARS) levels in the brain [43]. In traumatic brain injury models using male C57BL/6 mice, administration of 20 mg/kg per day reduced pro-inflammatory cytokine release, microglial activation, and cellular apoptosis, demonstrating neuroprotective effects [44]. Another study administering 30 mg/kg per day for 21 days highlighted crocin’s ability to counteract chronic stress-induced oxidative damage in the brain, kidneys, and liver of rats by modulating antioxidant enzyme levels [45]. While carotenoids protect against endogenous DNA damage in vivo, their protective effects against hydrogen peroxide-induced damage ex vivo are less pronounced, possibly due to reduced antioxidant interactions [46].

To summarize, carotenoids such as β-carotene, ASTA, and saffron exhibit anti-inflammatory and antioxidant effects, as shown in both in vitro and in vivo studies (Table 2). Evidence suggests these compounds may mitigate cytokine storms and pro-inflammatory effects, mainly through NF-κB and JAK/STAT signaling pathways, with potential benefits for autoimmune conditions [27]. However, human intervention studies on the anti-inflammatory effects of carotenoids have yielded mixed and inconclusive results [47].

**Table 2 antioxidants-14-01111-t002:** Studies that show the effects of different carotenoids that have a relation to the immune system.

Carotenoid	Type of Study	Model Details	Activity	Formulation/Treatment Details	References
β-carotene	In vitro	3T3-L1 preadipocytes	Reduces pro-inflammatory adipokines	5–10 µM (dissolved in tetrahydrofuran)	Cho et al., 2018 [27]
	In vitro	AGS cell line	Inhibits NF-κB activation	Synthetic; 20 µM (dissolved in tetrahydrofuran) Synthetic; 2–20 µM (dissolved in tetrahydrofuran)	Kim et al., 2011 [28]; Jang et al., 2009 [29]
	In vitro	AGS cell line	Reduces oxidative stress	Synthetic; 20 µmol/L (dissolved in tetrahydrofuran)	Kim et al., 2011 [28]
	In vitro	Murine macrophage cell line RAW264.7 and murine peritoneal macrophages	Reduces phosphorylation levels of JAK2 and STAT3	Synthetic; 10–100 µM	Li et al., 2019 [30]
	In vitro	Murine macrophage cell line RAW264.7 and murine peritoneal macrophages	Reduces phosphorylation levels of JNK and p38	Synthetic; 10–100 µM	Li et al., 2019 [30]
Astaxanthin (ASTA)	Clinical trial	Healthy volunteers and smokers	Decreases lipid peroxidation and protect against cellular damage	Algae-derived; Softgel capsules (3 mg) for 4 weeks; 5–40 mg for 3 weeks	Imai et al., 2018 [33]; Kim et al., 2011 [37]
	In vitro	THP-1 macrophage cells	Inhibits NF-κB activation	Synthetic; 5–10 µM (dissolved in dimethyl sulfoxide)	Kishimoto et al., 2010 [35]
	In vitro	THP-1 macrophage cells	Downregulates pro-inflammatory markers	Synthetic; 5–10 µM (dissolved in dimethyl sulfoxide)	Kishimoto et al., 2010 [35];
	In vitro	AGS cell line	Inhibits PI3K/AKT/mTOR signaling pathway	Synthetic; 1–10 µM (dissolved in dimethyl sulfoxide)	Utpal et al., 2024 [36]
Saffron (crocin and crocetin)	In vitro	PC12 cells	Increases intracellular glutathione, protecting from cell death	Plant-derived; 10 µM in serum-free DMEM	Ochiai et al., 2007 [39]
	In vivo	Sprague–Dawley male rats	Downregulates pro-inflammatory markers	Synthetic; 1 mL/kg (dissolved in normal saline)	Yang et al., 2006 [40]
	In vitro	BV2 mouse brain microglial cells	Reduces NF-κB activation and mitigate oxidative stress	Synthetic; 5–20 µM (dissolved in fresh medium)	Nam et al., 2010 [41]
	In vivo	Adult male Wistar rats and C57BL/6 mice	Neuroprotective potential	Synthetic; 20–60 mg/kg per day (dissolved in normal saline); between 3 and 6 weeks	Rajaei et al., 2016 [43]; Wang et al., 2015 [44]; Bandegi et al., 2014 [45]

### 2.2. Immunomodulatory Activity of Carotenoids on Cell Populations of the Immune System

The current literature focuses on elucidating the mechanisms through which carotenoids exert their antioxidant and anti-inflammatory properties. Although significant advances have been made, much remains to be understood. Research also focuses on the impact of carotenoids on various immune cell functions, including lymphocyte proliferation, cytokine release, phagocytic and microbicidal capacities, natural killer (NK) cell cytotoxicity, and inflammatory responses. These effects have been demonstrated through various in vitro and in vivo studies [48]. In this review, beyond compiling existing research on the antioxidant and immunomodulatory properties of different carotenoids, we also aim to summarize the most recent studies that focus on specific components and populations of the immune system, analyzing how the most-studied carotenoids influence each of these immune elements (Table 3). This targeted approach adds both novelty and relevance to the present review, offering a more detailed and updated perspective on the immunological roles of carotenoids.

#### 2.2.1. Peripheral Blood Mononuclear Cells

Regarding the effect of carotenoids on peripheral blood mononuclear cells (PBMCs), oral supplementation of β-carotene (0–60 mg/day) for 2 months can increase T helper lymphoid cells and NK cells, as well as their activation markers IL-2 and transferrin [49]. Additionally, synthetic β-carotene supplementation, at a concentration of 50 mg/day in elderly individuals, significantly improved NK cell function [50]. Also, carrot juice rich in β-carotene did not affect PBMC proliferation during supplementation, but one week after stopping, lymphocyte proliferation and NK lytic activity were significantly increased. Also, a gradual response of β-carotene has been demonstrated. In addition, carrot juice rich in β-carotene increased lymphocyte proliferation and NK lytic activity only after supplementation, suggesting a delayed effect [51]. Finally, a hormetic effect with 3 µM of synthetic β-carotene has been associated with reduced tumor-destroying capability of NK cells in vitro, whereas lower doses enhance this activity, and, while low concentrations of β-carotene do not affect erythrocyte or leukocyte viability, high concentrations significantly reduce viability [52].

ASTA, with its three functional isomers, has been shown to enhance lymphocyte proliferation and the phagocytic activity of peritoneal exudate cells [53], being the 3S, 3′S enantiomer the one with superior immunoregulatory potential [54]. Synthetic ASTA also shows no toxicity on lymphocytes, neither in vivo nor ex vivo, and enhances cell viability in LPS-induced proliferation with a supplementation of 7 mg/kg per day in BALB/c mice [55].

Curcumin promotes the proliferative capacity of T-cells and has no negative effect on ROS and NO production by macrophages and on NK cell cytotoxicity in curcumin-injected rats at a concentration of 40 mg/kg per day [56]. Further research demonstrates synthetic curcumin’s ability (0.2–2 µg/mL) to modulate T-cell-mediated inflammation by restraining CD4+ T-cell proliferation induced by CD2/CD3/CD28, regulating CD69, CCR7, L-selectin, and TGFβ1 expression depending on the phase of the T cell activation [57].

In vitro studies have shown that lycopene reduces T-cell activation and expression of IL-2, a key cytokine in T-cell stimulation [58]. Moreover, an in vivo study with supplementation with 40 mg/kg per day in C57BL/6 mice of lycopene shows an increase in the CD4+/CD8+ T-cell ratio and in the percentages of IFNγ+, perforin+, and granzyme B+ expression on CD8+ T cells and an enhanced apoptosis in tumor cells with combined lycopene and anti-PD-1 treatment in comparison with those of either lycopene or anti-PD-1 alone [59]. It also positively impacts NK cell survival and cytotoxicity, with its anti-apoptotic effects linked to reduced expression of caspase-3 and -9 genes. At last, in a randomized, double-blind study, daily supplementation with 25–50 mg of lycopene or α- and β-carotene over two weeks improved NK cell cytotoxicity and lymphocyte proliferation in healthy males [51].

Finally, retinoic acid (RA), the primary vitamin A metabolite, cooperates with catalase to induce monocyte differentiation into macrophages, and it can remove H_2_O_2_ effectively [60]. It has also been demonstrated that RA promotes immune tolerance by inducing the differentiation of highly suppressive regulatory T cells from peripheral blood mononuclear cells [61]. In recent years, RA has been widely investigated and applied in several studies for its anti-inflammatory properties. It has been shown to suppress Th1/Th17 immune responses after synthetic administration in C57BL/6 rats at a 0.2 mg concentration for 21 days [62]. Also, it has been reported that RA enhances the expansion of regulatory T cells while inhibiting the differentiation of Th17 cells [63].

#### 2.2.2. Polymorphonuclear Leukocytes

Polymorphonuclear leukocytes (PMNs) supplemented in vitro or in vivo with 1 µM of synthetic β-carotene normalized ROS content, modulating their amount during oxidative burst, probably by a quenching effect. In addition, β-carotene might act by modulating the activity of enzymes implicated in the ROS biochemical pathway [64].

ASTA has been shown to significantly enhance neutrophil phagocytic and microbicidal capabilities. This enhancement is accompanied by increased intracellular calcium (Ca^2+^) levels and nitric oxide (NO) production, as well as a reduction in superoxide anion and hydrogen peroxide (H_2_O_2_) levels and a decrease in pro-inflammatory cytokines IL-6 and TNF-α. Moreover, ASTA treatment reduces oxidative damage to proteins and lipids, underscoring its protective role against oxidative stress [53]. In a separate study, ASTA demonstrated no toxicity in human cells at low concentrations, although high concentrations of ASTA in the culture medium led to reduced neutrophil viability. Despite this, neutrophils treated with 0–40 µM of synthetic ASTA exhibited significant increases in both phagocytic and fungicidal capacities, highlighting its potential for immune enhancement [65].

Fucoxanthin (Fc) or Vitamin C (Vc) causes neutrophil death at any tested concentrations. Cell viability assays confirmed no significant loss in cell membrane integrity with Fc and Vc, whether administered alone or in combination. Although Fc alone or combined with Vc did not alter neutrophil migration in response to peptide N-formyl-methionyl-leucyl-phenylalanine (fMLP) stimulation, an increase in neutrophil phagocytic capacity was observed across all experimental groups with a supplementation of 2–100 µM [66]. Further results demonstrated increased intracellular calcium concentration and mobilization in neutrophils treated with Fc and Vc. Additionally, Fc and Vc treatment led to decreased production of TNF-α and superoxide anions, supporting their role in reducing pro-inflammatory responses and oxidative stress [66].

#### 2.2.3. Dendritic Cells

Curcumin has been shown to enhance the antigen-capturing abilities of dendritic cells (DCs) through a mannose receptor-mediated endocytosis mechanism. This enhancement is dose-dependent, varying based on curcumin concentration. In addition to promoting antigen capture, curcumin inhibits LPS-induced MAPK activation and NF-κBp65 translocation in DCs. It significantly downregulates the expression of costimulatory molecules CD80 and CD86, as well as MHC class II, though it does not affect MHC class I expression. Additionally, synthetic curcumin (0–25 µM) restricts the maturation of murine bone marrow-derived DCs [67]. Also, curcumin from *Curcuma longa* shows inhibitory effects on DC activity, revealing that it reduces cell migration and chemokine secretion [68]. Moreover, curcumin’s suppressive action on DC activation is mediated via the modulation of the JAK/STAT/SOCS signaling pathway [69].

The biosynthetic C_30_ carotenoid 4,4′-diaponeurosporene (Dia) has been shown to influence the morphology and maturation of DCs, promoting dendritic elongation and an increase in the shape index of DCs (a parameter used to describe and assess the morphology of these immune cells). Mature DCs play crucial roles in regulating immune responses by secreting a range of cytokines. Dia obtained from metabolic engineering of *Bacillus subtilis* enhances the production of both pro-inflammatory and anti-inflammatory cytokines in DCs obtained from C57BL/6 mice, including IL-6, IL-10, IL-12p70, and TNF-α, thereby supporting DC-mediated immune regulation and significantly upregulating CD40, CD80, CD86, and MHC class II expression [70].

#### 2.2.4. Effect on Cytokines

Curcumin has been shown to inhibit the production of Th1 cytokines, specifically IL-2 and IFN-γ, in both macrophages and splenic T lymphocytes pre-exposed to this compound at a concentration of 100 mM [71]. Dietary intake of curcumin and limonin has also demonstrated a suppressive effect on CD4+ T-cell proliferation and IL-2 production, as well as on the nuclear translocation of NF-κB p65 in activated CD4+ T-cells in DO11.10 transgenic mice [72].

Studies suggest that lycopene can influence the immune response, as indicated by its effects on pro-inflammatory cytokine production. Increased levels of IL-1β and TNF-α, along with reduced secretion of the anti-inflammatory cytokine IL-10, suggest that lycopene may promote inflammatory reactions [73]. However, other research demonstrates lycopene’s ability at a concentration of 2 µM to reduce pro-inflammatory cytokines and chemokines, including IL-6, IL-1β, and MCP-1, at both mRNA and protein levels in various adipose tissues and adipocyte models [74]. Further, lycopene has been shown to elevate levels of IFN-β, IFN-γ, interferon regulatory factor 1 (IRF1), IRF7, and C-X-C motif chemokine ligands 9 and 10 (CXCL9 and CXCL10), while reducing levels of IL-4, IL-10, DNA methyltransferase 3 alpha (DNMT3a), and methylation of IRF1 and IRF7 promoters, leading to a decrease in tumor volume in C57BL/6 mice after 40 mg/kg per day supplementation [59].

β-Carotene supplementation (0–10 µM) has demonstrated the ability to inhibit the transcription of cytokines such as IL-1β, IL-6, and IL-12 [75]. In another study, oral administration of β-carotene at a concentration of 5 mg/kg per day in C57BL/6 mice along with capsaicin led to a substantial increase in IFN-γ and IL-5 production, with levels up to three times higher than those observed with capsaicin alone [76].

The effects of ASTA on cytokine production appear to be context dependent. While ASTA had no significant effect on IFN-γ and IL-2 production in primary cultured lymphocytes, it enhanced IFN-γ production in response to LPS stimulation. Enzyme-linked immunosorbent assay (ELISA) results confirmed that astaxanthin significantly increased IFN-γ levels when LPS was used as a stimulus, whereas IL-2 production was only increased following Con A stimulation. Additionally, astaxanthin alone (7 mg/kg per day) significantly elevated IL-2 production in BALB/c mouse lymphocytes, although it did not significantly affect IFN-γ production [55].

**Table 3 antioxidants-14-01111-t003:** Studies that show some of the influence of the main carotenoids on the immune system cell populations.

Cellular Group	Carotenoid	Type of Study	Activity	Formulation/Treatment Details	References
PBMCs	β-carotene	Clinical trial	Increments lymphoid cells	0–60 mg/day; 2 months	Watson et al., 1991 [49]
		Clinical trial	Increases cytotoxic activity of NK cells	Synthetic; 50 mg/day for 2 years;	Santos et al., 1996 [50];
		Clinical trial	Increases cytotoxic activity of NK cells	Dietary source; 25 mg/day for 2 weeks	Watzl et al., 2003 [51]
		In vitro	Modulates the viability of immune cells	Synthetic; 0–3 µM (dissolved in acetone)	Ribeiro et al., 2020 [52]
	Astaxanthin	In vitro	Enhances phagocytic capabilities	Synthetic; 10 µM (dissolved in dimethyl sulfoxide); 20 μM	Speranza et al., 2012 [53]; Sun et al., 2016 [54]
		In vitro	Modulates lymphocytes proliferation	Synthetic; 10 µM (dissolved in dimethyl sulfoxide); 20 μmol/L; 70–300 nM)	Speranza et al., 2012 [53]; Sun et al., 2016 [54]; Lin et al., 2015 [55]
		Ex vivo (BALB/c mice)	Modulates lymphocytes proliferation	Synthetic; 7 mg/kg per day by oral gavage for 14 days	Lin et al., 2015 [55]
	Curcumin	In vivo (Wistar rats)	Promotes the proliferative capacity of T-cells	Synthetic; 40 mg/kg per day for 30 days (Injection)	Varalakshmi et al., 2008 [56]
		In vitro	Modulates T-cell-mediated inflammation	Synthetic; 0.2–2 µg/mL	Kim et al., 2013 [57]
	Lycopene	Clinical trial	Improves NK cytotoxicity	Dietary source; 25–50 mg/day for 2 weeks	Watzl et al., 2003 [51]
		In vitro	Reduces T-cell activation	Synthetic; 1–10 µM	Kim et al., 2004 [58]
		In vivo (C57BL/6 mice)	Increases the CD4+/CD8+ T-cell ratio	40 mg/kg per day (intraperitoneal injection)	Jiang et al., 2019 [59]
		In vitro	Induces macrophages differentiation	Synthetic; 0.1 µM	Ding et al., 2007 [60]
	Retinoic acid	In vitro	Induces regulatory T cells differentiation	Synthetic; 10–100 nM (dissolved in dimethyl sulfoxide);	Wang et al., 2009 [61]; Xiao et al., 2008 [63]
		In vivo (C57BL/6 mice)	Suppresses Th1/Th17 immune responses	Synthetic; 0.2 mg/mouse per day for 21 days (intraperitoneally)	Keino et al., 2010 [62]
PMNs	β -carotene	In vitro	Modulates ROS content	Synthetic; 1 µM (dissolved in tetrahydrofuran)	Walrand et al., 2005 [64]
	Astaxanthin	In vitro	Enhances neutrophil phagocytic and microbicidal capabilities	Synthetic; 0–40 µM (dissolved in dimethyl sulfoxide)	Macedo et al., 2010 [65]
	Fucoxanthin and vitamin C	In vitro	Modulates neutrophil phagocytic capacity and viability	Synthetic; 2–100 µM (dissolved in dimethyl sulfoxide)	Morandi et al., 2014 [66]
Dendritic cells	Curcumin	In vitro (from murine BM cells)	Restricts the maturation of DCs	Synthetic; 0–25 µM (dissolved in dimethyl sulfoxide)	Kim et al., 2005 [67]
		In vitro	Suppresses activation of DCs	Plant-derived; 20–30 µM (dissolved in dimethyl sulfoxide)	Shirley et al., 2008 [68]
		In vivo (C57BL/6 mice)	Suppresses activation of DCs	Synthetic; 100 mg/kg; oral gavage for 7 days	Zhao et al., 2016 [69]
	Dia	In vitro (from C57BL/6 mice)	Influences morphology and maturation	Bacterial-derived; 1 µM	Liu et al., 2016 [70]
Cytokines	Curcumin	In vitro	Inhibits Th1 cytokines	Synthetic; 100 mM (dissolved in dimethyl sulfoxide)	Gao et al., 2004 [71]
		In vitro	Inhibits Th1 cytokines	Dietary source; 1 g/100 g	Kim et al., 2009 [72]
	Lycopene	In vitro	Modulates the production of pro- and anti-inflammatory cytokines	Dietary source; 0.25–4 µM	Bessler et al., 2008 [73]
		In vitro		Synthetic; 2 µM	Gouranton et al., 2011 [74]
		In vivo (C57BL/6 mice)		40 mg/kg per day (intraperitoneal injection)	Jiang et al., 2019 [59]
	ß-carotene	In vitro (RAW264 cells)	Inhibits pro-inflammatory cytokines	Synthetic; 0–10 µM	Katsuura et al., 2009 [75]
		In vivo (C57BL/6N mice)	Modulates Th cytokine production	Synthetic; 5 mg/kg per day for 7 days	Yamaguchi et al., 2010 [76]
	Astaxanthin	Ex vivo (BALB/c mice)	Modulates IFN-γ and IL-2 production	Synthetic; 7 mg/kg per day by oral gavage for 14 days	Lin et al., 2015 [55]

## 3. Relationship Between Carotenoid Intake and Prevention of Diseases

### 3.1. Role of Carotenoids in Autoimmune Diseases

Autoimmune diseases (ADs) are complex, multifactorial disorders characterized by loss of immune self-tolerance, leading to immune attacks on host tissues and consequent organ damage [77]. They are highly prevalent, particularly among young individuals, and impose substantial medical costs. ADs differ in their clinical manifestations, affecting single or multiple organs, as seen in multiple sclerosis (MS) and type 1 diabetes (T1D), where immune responses initially target oligodendrocytes and pancreatic β-cells, respectively, but may later extend to other tissues [78]. Both genetic predisposition and environmental exposures contribute to AD pathogenesis by disrupting T- and B-cell regulation, with environmental agents capable of activating autoreactive cells. Impaired efferocytosis, the defective clearance of apoptotic cells, is another hallmark of ADs. Natural products, due to their antioxidant and anti-inflammatory properties, are emerging as promising therapeutic strategies, offering efficacy with low toxicity and growing evidence of potential as adjunctive treatments (Table 4) [79].

#### 3.1.1. Multiple Sclerosis

Multiple Sclerosis (MS) is a chronic autoimmune and neuroinflammatory disease characterized by immune-mediated damage to the central nervous system (CNS) [80]. The pathophysiology of MS is complex, involving both genetic susceptibility and environmental triggers that disrupt immune tolerance to CNS antigens. One potential mechanism linked to the disease’s progression involves microbiota imbalance, which may foster a pro-inflammatory systemic environment, thereby worsening disease severity [81,82]. Emerging studies suggest that the dysbiosis of gut microbiota can influence immune function and may play a critical role in MS development by enhancing systemic and CNS inflammation, which are hallmarks of the disease.

The disease pathology in MS is largely driven by two principal features: acute inflammation and demyelination. These features result from immune cell infiltration into the CNS, which leads to damage in the myelin sheath, neuronal degeneration, and secondary axonal damage. Syncytin-1, a protein known to activate pro-inflammatory pathways, has been identified as a trigger for these autoimmune cascades, amplifying immune responses against CNS tissues and exacerbating disease pathology [83]. Given the inflammatory component of MS, several natural compounds have shown promise in mitigating these processes. For instance, saffron has exhibited protective effects in MS models, such as experimental autoimmune encephalomyelitis (EAE) in mice, where it effectively reduces leukocyte infiltration into the CNS and curbs oxidative stress. These effects have been attributed to crocin, a bioactive compound in saffron, which may reduce neuroinflammation and provide symptomatic relief in MS. Crocin’s additional antidepressant properties could be particularly beneficial given the high incidence of depressive symptoms in MS patients, and it has shown anti-neuroinflammatory properties with 500 mg/kg supplementation for 21 days in mice [84].

Recent randomized controlled trials have explored the potential of other natural compounds, such as lutein, in modulating biological markers relevant to MS. One clinical trial observed that lutein supplementation (20 mg per day) increased macular pigmentation, skin carotenoid levels, and serum lutein concentrations in individuals with MS. Although there was no immediate impact on cognitive function, increased macular pigmentation was linked to improvements in attentional inhibition and spatial memory, suggesting an indirect cognitive benefit [85]. A novel diagnostic approach using Raman spectroscopy has highlighted the potential of lipid and carotenoid molecules as biomarkers for MS. This technique identified specific lipid and carotenoid alterations that distinguished MS samples from healthy controls, particularly through reduced carotenoid levels and lipid profile changes, offering potential applications in early diagnosis and treatment monitoring [86].

#### 3.1.2. Systemic Lupus Erythematosus

Systemic Lupus Erythematosus (SLE) is a chronic autoimmune disorder characterized by the production of autoantibodies that target nuclear and cytoplasmic components of cells. This self-directed immune response causes widespread tissue damage and systemic inflammation [87,88]. SLE can affect multiple organ systems, including the cardiovascular, renal, musculoskeletal, and nervous systems, resulting in diverse and often severe clinical manifestations. Autoantibody production in SLE is believed to be triggered by molecular mimicry, abnormal antigen presentation, and modification of endogenous antigens, which lead to the formation of pathogenic immune complexes that deposit in tissues and drive inflammation [89].

In terms of natural interventions, lycopene has been studied for its impact on SLE. A study examining mortality in SLE patients found that those with higher serum levels of lycopene had a significantly lower mortality rate (7.1%) than those with lower levels (37.5%). The protective effect was even more pronounced in individuals over 50 years old, with a mortality rate of 0% in the high-lycopene group versus 71.4% in the low-lycopene group [90]. Furthermore, adherence to a Mediterranean diet rich in β-carotene has been linked to reductions in specific inflammatory biomarkers, such as high-sensitivity C-reactive protein (hsCRP) and homocysteine (Hcy), both of which are associated with increased cardiovascular risk in SLE patients [91]. These findings underscore the importance of dietary antioxidants in managing systemic inflammation and cardiovascular risk in autoimmune diseases like SLE.

#### 3.1.3. Rheumatoid Arthritis

Rheumatoid Arthritis (RA) is a systemic autoimmune disorder characterized by chronic inflammation of the synovial membranes and progressive erosion of articular cartilage. Affecting approximately 1–2% of the global population, RA disproportionately impacts women and often manifests between the ages of 40 and 60, though juvenile forms also occur [92]. Despite extensive research, the precise mechanisms underlying RA remain incompletely understood. However, studies indicate that pro-inflammatory cytokines, including tumor TNF-α and IL-1β, play central roles in RA pathogenesis by promoting synovial inflammation, cartilage degradation, and bone erosion [93].

The anti-inflammatory properties of carotenoids have led to interest in their potential as RA therapies. Observational studies consistently show that individuals with RA exhibit lower plasma levels of antioxidants, including carotenoids, compared to healthy controls, suggesting a potential protective role for these compounds in RA pathogenesis [94,95]. Crocin, a carotenoid extracted from saffron, has demonstrated significant anti-inflammatory effects in RA models. Animal studies show that crocin, at concentrations of 10–40 mg/kg per day, alleviates RA symptoms by reducing inflammation, decreasing chondrocyte death, mitigating joint damage, and lowering paw swelling. These effects occur in a dose-dependent manner and are accompanied by reductions in matrix metalloproteinases (MMP-1, MMP-3, and MMP-13) and pro-inflammatory cytokines such as TNF-α, IL-6, CXCL8, and IL-17, all of which are critical mediators of RA pathology [96].

In collagen-induced arthritis (CIA) mice, crocin treatment led to lower plasma levels of IL-1β, IL-6, and TNF-α, providing further evidence for its potential as an anti-inflammatory therapy [97]. A separate study found that crocin supplementation (10–20 mg/kg per day) in Wistar rats significantly reduced serum levels of inflammatory markers IL-1β, IL-6, TNF-α, and COX-2 in arthritic rats, and decreased ROS levels in serum, liver, and spleen by 98%, 96.5%, and 98%, respectively. Crocin also enhanced antioxidant defense mechanisms by increasing levels of catalase (CAT), glutathione (GSH), glutathione S-transferase (GST), and superoxide dismutase (SOD) [98].

In clinical contexts, lycopene, another carotenoid, has also shown promise for RA management. A nanopharmaceutical formulation of lycopene was found to reduce knee joint thickness and immune cell infiltration in RA patients, highlighting the enhanced efficacy of nanoformulated lycopene compared to pure lycopene at a 1 mg/kg administration in C57BL/6 mice [99]. Additionally, epidemiological studies have linked higher dietary intake of β-cryptoxanthin and zeaxanthin with a reduced risk of inflammatory polyarthritis, suggesting that dietary antioxidants could offer protective benefits against RA [100].

**Table 4 antioxidants-14-01111-t004:** Role of carotenoids in the most common autoimmune diseases.

Autoimmune Disease	Carotenoid	Type of Study	Activity	Formulation/Treatment Details	References
Multiple sclerosis	Crocin	In vivo (C57BL/6 mice)	Anti-neuroinflammatory and antidepressant properties	Plant-derived; 500 mg/kg per day for 21 days	Ghazavi et al., 2009 [84]
	Lutein	Clinical trial	Cognitive benefits	Plant-derived; 20 mg per day for 4 months	Martell et al., 2023 [85]
Systemic lupus erythematosus	β-carotene	Cross-sectional study	Reduces specific inflammatory biomarkers	Dietary source; 6 months	Pocovi-Gerardino et al., 2021 [91]
Rheumatoid arthritis	Crocin	In vivo (Wistar rats)	Reduces pro-inflammatory cytokines	10–40 mg/kg; oral treatment for 36 days	Liu et al., 2018 [96]
		In vivo (Wistar rats)	Reduces ROS serum levels	Synthetic; 10–20 mg/kg; oral treatment for 2 weeks	Hemshekhar et al., 2012 [98]
	Lycopene	In vivo (C57BL/6 mice)	Inhibits the infiltration of leukocytes, mononuclear cells and neutrophils at the site of inflammation	Pure lycopene or nanoemulsion (1 mg/kg)	Moia et al., 2020 [99]

### 3.2. Role of Carotenoids in Other Types of Diseases

Carotenoids are recognized for their potent antioxidant properties and, given their ability to neutralize reactive oxygen species and reduce oxidative stress, considerable research has been conducted to evaluate the influence of carotenoid intake on the risk and progression of various diseases, including cancer and other significant medical conditions (Table 5) [79].

#### 3.2.1. Cancer

Extensive investigations have shed light on the potential protective role of carotenoids against cancer development. Epidemiological studies have demonstrated that higher circulating levels are linked to reduced breast cancer risk. In vitro, lycopene and β-carotene inhibit proliferation, induce cell cycle arrest, and promote apoptosis in breast cancer cells, highlighting their role in regulating cell growth and survival [101,102].

Moreover, the effects of β-carotene on hematological malignancies have also been explored. In studies with human chronic monocytic leukemia (U937) and myeloid leukemia cell lines, supplementation with 0.5–10 µM of synthetic β-carotene demonstrated a dual role: it acted as an antioxidant at lower concentrations and exhibiting prooxidant activity at higher concentrations [102].

Saffron from *Crocus sativus* exhibits anticancer effects by reducing HepG2 cell viability in a time- and dose-dependent manner. It downregulates TNF-R1, COX-2, iNOS, and NF-κB-p65, while activating caspase-3, thereby inhibiting proliferation, lowering oxidative stress, and promoting apoptosis. In vivo, saffron administered at concentrations of 75–100 mg/kg for 22 weeks demonstrated anti-inflammatory and pro-apoptotic activity relevant to liver cancer therapy [103].

Lycopene has been extensively studied for its role in prostate cancer prevention. The prostate gland accumulates relatively high levels of lycopene, which may contribute to its protective effects. Several observational studies suggest an inverse relationship between dietary lycopene intake and the risk of prostate cancer. However, it is crucial to note that findings are not entirely consistent across all studies, and further research is needed to confirm lycopene’s efficacy and elucidate the underlying mechanisms [104].

#### 3.2.2. Cardiovascular Diseases

Cardiovascular diseases (CVD) are a leading cause of mortality worldwide, influenced by oxidative stress, chronic inflammation, dyslipidemia, and thrombogenesis. Carotenoids, recognized for their antioxidant properties, may exert protective effects against these factors [105,106]. Specifically, astaxanthin and lutein contribute to cardiovascular health by modulating lipid oxidation: astaxanthin delays LDL cholesterol oxidation, stabilizes LDL particles, and reduces their atherogenic potential, which may help prevent myocardial injury and other cardiac conditions [107,108].

Additionally, because β-carotene and lycopene are predominantly carried in LDL particles, they are strategically positioned to provide antioxidant protection against LDL oxidation. This protective role is crucial, as oxidized LDL is a major contributor to the development and progression of atherosclerosis and coronary heart disease [109].

The cardioprotective potential of carotenoids, and, in particular, lycopene, is also supported by large-scale analyses. A recent meta-analysis reviewed multiple studies and reported that individuals with higher dietary lycopene intake experienced a 17% reduction in the risk of cardiovascular disease. These findings underscore the potential of lycopene as a dietary intervention for cardiovascular risk reduction [110].

Carotenoids are also implicated in the prevention of atherosclerosis through direct mechanisms that, apart from inhibiting LDL oxidation, are protecting vascular endothelial cells from oxidative injury and preserving overall vascular function. These effects are instrumental in maintaining the integrity of the vascular system and preventing the formation of atherosclerotic plaques [111].

#### 3.2.3. Dermatological Diseases

Carotenoids, especially provitamin A types, benefit skin health through their conversion to retinoic acid, which regulates keratinocyte proliferation, epidermal differentiation, keratinization, inflammation, and oxidative stress, while enhancing absorption of topical substances [112]. Human studies show that UV exposure decreases carotenoid levels in plasma and skin, with lycopene being particularly susceptible, highlighting the importance of dietary intake or supplementation for skin protection. In a study with lycopene supplementation at a concentration of 120 mg for 6 days, it showed protection against UV damage [113].

Research investigating the photoprotective properties of carotenoids has yielded promising results. For example, supplementation with β-carotene has been shown to significantly reduce erythema induced by UV exposure from a solar simulator. The absence of carotenoid supplementation led to a marked increase in erythema formation, highlighting the protective effects of these antioxidants [114,115]. β-Carotene has revealed photoprotective properties against damage induced by visible and infrared radiation. Additionally, it can function as an effective antioxidant component in sunscreen formulations, offering protection against the adverse effects of UV radiation [116].

Studies on carotenoids such as β-carotene and canthaxanthin reveal explicit photoprotective properties. Experimental findings indicate that these carotenoids, through their ability to quench reactive oxygen species and other free radicals, provide substantial protection in patients with erythropoietic protoporphyria, a condition characterized by heightened photosensitivity. Notably, individuals with erythropoietic protoporphyria have been shown to have lower serum β-carotene levels and may require dietary supplementation to mitigate symptoms [2]. Furthermore, β-carotene and lycopene (0–8 µM) have been observed to decrease skin redness and damage following UV exposure, acting as soothing agents under intense sunlight [117].

Dietary interventions involving carotenoid-rich foods have also been associated with observable changes in skin coloration, indicative of carotenoid absorption and distribution within the skin layers. Among the carotenoids, ASTA has demonstrated pronounced benefits for skin protection, including reducing erythema and improving skin texture by minimizing wrinkles. Its strong antioxidant and anti-inflammatory properties are believed to contribute to these effects, interfering with UVA-induced MMP-1 and SFE/NEP expression after supplementation with synthetic ASTA (1–8 µM) [118]. A double-blind, placebo-controlled clinical study involved 24 healthy volunteers who received dietary carotenoid supplementation for eight weeks. The study results showed a significant increase in the skin’s radical scavenging activity, which conferred protection against reactive oxygen species (ROS) induced by environmental stressors. This evidence supports the role of dietary carotenoids in enhancing the skin’s natural defenses [119]. Astaxanthin has emerged as a potent anti-photoaging agent. It not only inhibits oxidative damage but also has anti-inflammatory and skin-lightening effects. Carotenoids, as dietary antioxidants, present a natural strategy for bolstering the skin’s resilience against photodamage and other environmental aggressors.

Lutein, another dietary carotenoid, has demonstrated significant skin benefits when administered orally. It has been shown to elevate antioxidant levels, reduce UV-induced erythema, enhance skin hydration, and improve skin elasticity [120].

#### 3.2.4. Ophthalmological Diseases

The human retina, particularly in the macula region, is rich in two essential dietary carotenoids: lutein and zeaxanthin, along with the isomer meso-zeaxanthin. These carotenoids are collectively known as macular pigment and are crucial for maintaining visual function. As the global population ages, age-related macular degeneration (AMD) is becoming an increasingly significant public health concern [121].

Research on the potential of lutein and zeaxanthin supplementation to mitigate the risk of AMD has produced mixed findings. While some studies have demonstrated a protective effect, others have not observed significant benefits. However, the variability in results may be partly due to limitations in study design and sample size, as many investigations were not adequately powered to yield conclusive outcomes [23]. Despite these inconsistencies, secondary analyses of data from the comprehensive Age-Related Eye Disease Study 2 (AREDS2) have provided valuable insights. The AREDS2 findings suggest that supplementation of these carotenoids at concentrations of 10 mg and 2 mg may have a role in slowing the progression of AMD in certain populations [122,123].

There is also growing evidence suggesting a link between lutein and zeaxanthin levels and the risk of developing nuclear cataracts. The AREDS2 trial revealed that individuals with lower dietary intake of these carotenoids experienced a reduced risk of progressing to cataract surgery when given lutein and zeaxanthin supplements. Specifically, those in the lowest quintile of dietary carotenoid intake showed a hazard ratio of 0.68 (95% CI: 0.48–0.96, *p* = 0.03), highlighting a potential protective effect [122]. These findings align with earlier observations by Yeum et al., who identified lutein and zeaxanthin as the sole carotenoids present in the human crystalline lens, suggesting their critical role in lens health [124].

#### 3.2.5. Neurological Diseases

Calcium (Ca^2+^) plays a vital role in intracellular signaling, and disruptions in Ca^2+^ signaling are implicated in the pathogenesis of several diseases. Carotenoids, such as astaxanthin, β-carotene, and lycopene, have been shown to influence Ca^2+^ ion transportation in the brain. Proper dietary intake of these carotenoids may help mitigate signaling malfunctions, reducing the risk of associated disorders [125].

Additionally, lycopene has been observed to influence the permeability of the blood–brain barrier, which may be compromised in certain neurological conditions [126]. Furthermore, a 5 mg/kg of lutein supplementation has been associated with a decreased risk of neurological diseases and a slower rate of cognitive decline [127].

Carotenoids’ antioxidant capabilities play a crucial role in protecting neural tissues from oxidative stress, a major contributor to neurodegeneration. Diets rich in carotenoids may also promote telomere health, as they help manage oxidative stress and regulate telomere length. By performing so, carotenoid-rich diets can contribute to preventing age-related neurological disorders, including Alzheimer’s disease [128].

**Table 5 antioxidants-14-01111-t005:** Role of carotenoids in other types of diseases.

Other Diseases	Carotenoid	Type of Study	Activity	Formulation/Treatment Details	References
Cancer	β-carotene	In vitro	Inhibits cell growth, halts the cell cycle at various stages, and promotes apoptosis Has anti- and pro-oxidant properties	Synthetic; 0.5–10 µM	Gloria et al., 2014 [102]
	Saffron	In vivo	Reduces liver cancer cell viability	Plant-derived; 75–300 mg/kg; orally for 22 weeks	Amin et al., 2011 [103]
Cardiovascular diseases	β-carotene	In vitro	Prevents LDL oxidation	Synthetic	Goulinet & Chapman, 1997 [109]
Dermatological diseases	Lycopene	Clinical trial	Protects against UV radiation damage	120 mg per day for 6 days	Ribaya-Mercado et al., 1995 [113]
	β-carotene	Clinical trial	Reduces erythema induced by UV exposure	Algae-derived; 25 mg capsules for 12 weeks	Stahl et al., 2000 [114]; Heinrich et al., 2003 [115]
		In vitro	Protects against UV radiation damage	Synthetic; 0–8 µM	Astley et al., 2003 [117]
	Astaxanthin	In vitro	Interferes with UVA-induced MMP-1 and SFE/NEP expression	Synthetic; 1–8 µM	Suganuma et al., 2010 [118];
Ophthalmological diseases	Lutein/Zeaxanthin	Clinical trial	Slows the progression of age-related macular degeneration	10 mg/2 mg daily	Chew et al., 2013 [122]
Neurological diseases	Lutein	In vivo	Decreases the risk of neurological diseases and slows the rate of cognitive decline	5 mg/kg per day (injected) for 21 days	Cho et al., 2018 [127]

## 4. Challenges in Carotenoid Research and Emerging C_50_ Carotenoids

Despite the promising evidence regarding the biological roles and therapeutic potential of carotenoids, several limitations and critical considerations must be acknowledged. One of the most significant challenges is the heterogeneity of study designs, which hampers the ability to compare outcomes across investigations. Preclinical studies, often conducted in cell lines or animal models, provide valuable mechanistic insights but may not fully capture the complexity of human physiology. Translational discrepancies between preclinical findings and clinical outcomes raise questions about the true efficacy of carotenoids in human health.

Another major limitation is the lack of standardized protocols for carotenoid intake or supplementation. Studies often differ in terms of the source (dietary, synthetic, or extract-based), formulation, and dosage, making it difficult to establish clear dose–response relationships. These factors contribute to inconsistencies across clinical trials and complicate the interpretation of results. Methodological issues also deserve consideration: small sample sizes, short intervention durations, and reliance on surrogate biomarkers rather than hard clinical endpoints limit the generalizability of many findings. Additionally, dietary assessment tools used in observational studies are prone to recall bias and measurement error, which may distort associations between carotenoid intake and disease outcomes.

Finally, an underexplored but critical aspect is the dual biological activity of carotenoids. They can act as antioxidants under certain conditions but may exert pro-oxidant effects at high concentrations or in specific cellular environments. This paradox highlights the need for carefully designed studies that account for context-dependent effects rather than assuming uniform benefits. Collectively, these limitations emphasize the importance of conducting well-powered, long-term randomized controlled trials with standardized supplementation protocols and rigorous endpoints to accurately define the role of carotenoids in human health and disease prevention.

As a final point, in addition to the variability observed in studies of common carotenoids, long-chain C_50_ carotenoids represent an emerging area that further contributes to the overall heterogeneity in carotenoid research. While most studies have focused on common carotenoids such as β-carotene, lutein, or astaxanthin, C_50_ carotenoids have gained increasing attention in recent years due to their unique structures and potential biological properties [129]. C_50_ carotenoids exhibit strong antioxidant properties due to their long-conjugated double-bond chains and the presence of multiple -OH groups. These structural features enhance their capacity to scavenge free radicals and protect biomolecules from oxidation, demonstrating superior antioxidant activity compared to commonly studied carotenoids, even at much lower concentrations [130]. Among them, bacterioruberin stands out for its strong antioxidant capacity and possible immunomodulatory effects, although the available evidence remains limited. Notably, bacterioruberin has been investigated in cancer models, showing promising results that warrant additional studies [131,132]. These emerging compounds represent a promising area for future research in the context of immune health and diseases related to oxidative stress.

## 5. Conclusions

Based on the evidence collected through all these studies, it can be conclusively stated that carotenoids play a fundamental role in regulating and strengthening the immune system. Its beneficial effects cover a wide range of immunological functions, from improving the activity of macrophages and lymphocytes to regulating the adaptive immune response. The mechanisms by which carotenoids exert their protective influence are diverse and complex, including the modulation of cytokine expression, regulation of cell differentiation and proliferation, and reduction in oxidative stress.

Furthermore, epidemiological and intervention studies have revealed the association between greater consumption of foods rich in carotenoids and a lower incidence of infectious and chronic diseases, including certain types of cancer, cardiovascular diseases, or autoimmune diseases [20,79]. Evidence suggests that carotenoids may attenuate the progression of autoimmune diseases by modulating the overactive immune response and suppressing autoantibody production.

In recent years, the discovery of C_50_ carotenoids has sparked significant interest due to their remarkable properties, which surpass those of more common carotenoids. These unique compounds, like bacterioruberin, exhibit enhanced antioxidant capabilities and increased stability, making them a promising area of research. Notably, their potential role in immune system modulation has drawn attention, as preliminary studies suggest they could influence immune responses more effectively [131]. This opens new avenues for exploring their application in dietary supplements or therapeutic strategies aimed at improving immune health.

However, although the literature convincingly supports the immunomodulatory benefits of carotenoids, gaps remain in our understanding of the specific molecular mechanisms underlying their effects. Likewise, the optimization of carotenoid intake patterns remains a relevant issue since their bioavailability and interactions with other nutrients could significantly influence their clinical efficacy.

Together, this research underscores the importance of considering carotenoids as essential components in promoting a balanced immune response and preventing disease, supporting the need for further studies to fully understand their potential therapeutic effects and to be aware of the prevention, development, and treatment strategies focused on immune health. Also, there is a growing interest in exploring less-common carotenoids, especially those with longer carbon chains, as they are increasingly demonstrating superior bioactivity compared to the more extensively studied shorter-chain carotenoids, highlighting the need to expand current research efforts to fully uncover their distinct biological roles and potential clinical applications.

## Figures and Tables

**Figure 1 antioxidants-14-01111-f001:**
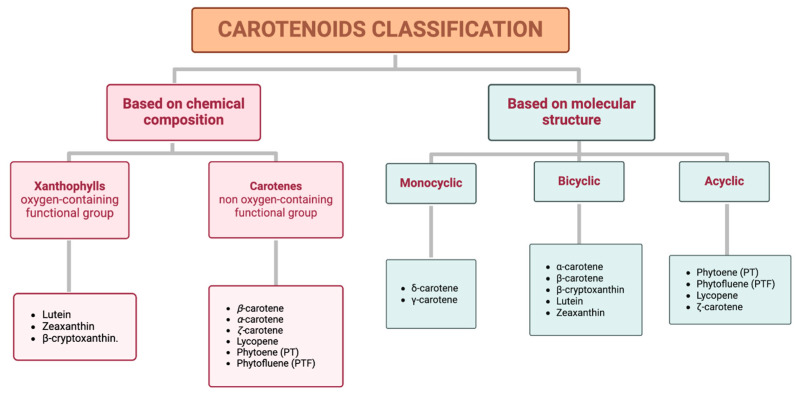
Classification of carotenoids based on their chemical composition (xanthophylls and carotenes) and molecular structure (monocyclic, bicyclic, and acyclic).

**Figure 2 antioxidants-14-01111-f002:**
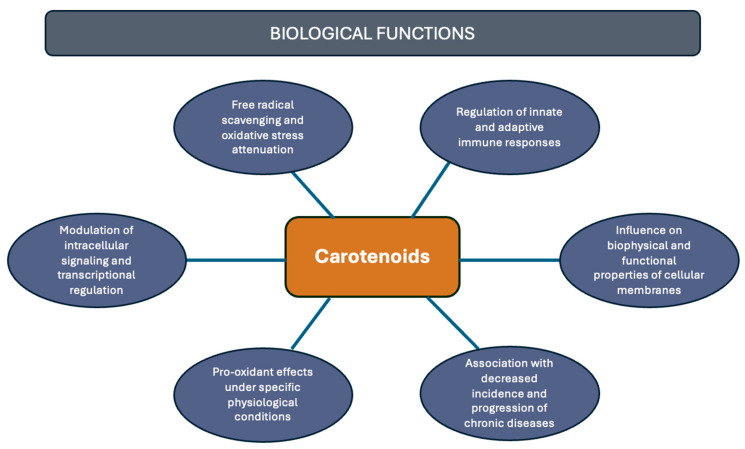
Health-related functions of carotenoids.

## Data Availability

No data were created in this study. Data sharing does not apply to this article.

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
