# Peer review of "Carotenoids and Their Interaction with the Immune System"

_antioxidants, 2025, doi:10.3390/antiox14091111_

Round 1
Reviewer 1 Report
The main goal of present review is to summarize in detailrecent data regarding special effects of carotenoids in immune system. Although carotenoids like β-carotene and α-carotene were primarily recognized for being a provitamin A source, essential for vision and immune system, many other effects were described and tested both in vitro and in vivo studies. Recent scientific evidences have revealed a multitude of potential health benefits and a reduced risk of various diseases associated with the consumption of carotenoids. Except their antioxidant capacity, other properties were highlighted, including regulation of intracellular signaling pathways, light absorption, modulation of membrane properties, and other various aspects associated with immune system modulation. Thus, recent and detailed discussion of relationship between carotenoid presence and the incidence of various health conditions is the main topic of present study. The data are up to date and the review can be considered as useful summary of recent data regarding biological effect of carotenoids.
Nevertheless, there are some open questions and problems to discussion before publication of this study. The main notes are introduced below.
After major revision I can imagine that this review could be accepted for publication in the Antioxidants Journal.
Formal notes:
line 47 - correct is "non-photosynthetic..."
line 139 - Table 1 is referred instead to Table 2
Conceptualization notes:
- Chapter 2.1 is too heterogenous; it is not clear if the effect deccribed is related to individual compound, or it was received as a result of in vitro study on cell lines or in vivo trial after supplementation by some (what?) carotenoid. I recommend to separate this chapter to 3 sub-chapters focused separately on above mentioned parts.
- In overview of effects of individual carotenoids with the immune system should be introduced the origin, source (plant, algae, microorganisms) and type of carotenoid preparative (complex natural mixture; isolated/purified chemical "individuum", synthetic preparative). It is also necessary to intruduce used dose of applied carotenoids.
- The data regarding type of carotenoid preparative and the the realization and arrangement of a study should be added into Tables 2, 3 and 4.
- The range of chapters is unbalanced; chapter 3 is too extensive and contains a large number of inconsistent data regarding the "real" effect of carotenoids on various diseases; a combination of serious clinical trials and simple monitoring of blood levels of selected parameters can lead to misleading interpretation. I recommend shortening the Chapter 3, to revise and give only selected substantial studies to Table 4 so that carotenoids are not considered to be a cure for most of the serious diseases. Moreover, there is no recommended optimum daily dose.
Reviewer 2 Report
In the manuscript authored by Miguel Medina-García et al., the primary aim is to summarize the health benefits of carotenoids. While the review provides valuable information, it would benefit from a more critical perspective. Currently, it emphasizes positive outcomes without adequately addressing the limitations and challenges of carotenoid research, including heterogeneity of study designs, variability in bioavailability, and gaps in knowledge regarding dosing, safety, and clinical translation. Incorporating these aspects would significantly enhance the rigor and relevance of the manuscript.
Despite the valuable contribution, several aspects could be improved to enhance clarity, scientific rigor, and completeness.
-Figure 2 presents a very simplistic view of the beneficial effects of carotenoids. Phrases such as “reduce risk of some disease” are too broad and do not convey precise biological or clinical mechanisms. I suggest refining the figure to clearly depict specific effects authors intend to mentioned in the text
-summary tables are currently incomplete. To provide a thorough and useful overview, I recommend including key details for each study: the type and source of carotenoids, dosage, study type (clinical or preclinical), model details (species, strain, gender), treatment duration, and relevant outcomes. This would greatly increase the reproducibility and practical utility of the tables.
- in the discussion, the authors mention C50 carotenoids only briefly in the conclusion, whereas earlier sections focus primarily on other carotenoids. To improve coherence and impact, the discussion should formally integrate C50 carotenoids, clearly highlighting their relevance and potential mechanisms of action. Notably, the authors introduce bacterioruberin as a particularly promising C50 carotenoid in the conclusion, yet no background or discussion about this compound is provided earlier in the text.
Several missing references in the discussion are present as such at line 601: “Furthermore, epidemiological and intervention studies have revealed the association between greater consumption of foods rich in carotenoids and a lower incidence of infectious and chronic diseases, including certain types of cancer, cardiovascular diseases, or autoimmune diseases.” Appropriate citation should be added to support these statement.
-A critical perspective is missing. I strongly recommend authors to include it in a separate chapter highlighting heterogeneity in study designs and outcomes, challenges in standardizing carotenoid intake or supplementation differences between preclinical and clinical results, and also to methodological limitations.
Round 2
Reviewer 2 Report
Authors adeguatelly reply to reviewer's comments and suggestions. The revised version of the manuscript has been improved and now it is suitable for publishing.
Authors adeguatelly reply to reviewer's comments and suggestions. The revised version of the manuscript has been improved and now it is suitable for publishing.
Author Response
Comments: Authors adeguatelly reply to reviewer's comments and suggestions. The revised version of the manuscript has been improved and now it is suitable for publishing.
Response: Thank you for your positive evaluation of the revised manuscript and for recognizing the improvements made. We appreciate your feedback during the review process.